# Molecular Pathogenesis of Human Immunodeficiency Virus-Associated Disease of Oropharyngeal Mucosal Epithelium

**DOI:** 10.3390/biomedicines11051444

**Published:** 2023-05-14

**Authors:** Sharof M. Tugizov

**Affiliations:** Department of Medicine, School of Medicine, University of California, San Francisco, CA 94143, USA; sharof.tugizov@ucsf.edu; Tel.: +1-(415)-514-3177

**Keywords:** human immunodeficiency virus, oropharyngeal mucosal epithelium, reactivation of opportunistic infections, disruption of epithelial junctions

## Abstract

The oropharyngeal mucosal epithelia have a polarized organization, which is critical for maintaining a highly efficient barrier as well as innate immune functions. In human immunodeficiency virus (HIV)/acquired immune deficiency syndrome (AIDS) disease, the barrier and innate immune functions of the oral mucosa are impaired via a number of mechanisms. The goal of this review was to discuss the molecular mechanisms of HIV/AIDS-associated changes in the oropharyngeal mucosa and their role in promoting HIV transmission and disease pathogenesis, notably the development of opportunistic infections, including human cytomegalovirus, herpes simplex virus, and Epstein-Barr virus. In addition, the significance of adult and newborn/infant oral mucosa in HIV resistance and transmission was analyzed. HIV/AIDS-associated changes in the oropharyngeal mucosal epithelium and their role in promoting human papillomavirus-positive and negative neoplastic malignancy are also discussed.

## 1. Introduction

Four decades ago, the HIV/AIDS pandemic began. Global spread led to 75 million infections and 32 million deaths. Today, highly effective prevention strategies are available to reduce the likelihood of HIV transmission via sexual intercourse. Furthermore, antiretroviral therapy for people living with HIV/AIDS can reduce viral loads to levels that cannot be detected or transmitted. However, despite the availability of these highly effective agents, HIV/AIDS continues to be a lethal disease. HIV/AIDS was responsible for one death each minute in 2021; every two minutes, a young woman becomes newly infected with HIV. Likewise, approximately 200,000 cases of mother-to-child transmission are reported each year. Thus, HIV/AIDS remains an unsolved problem; additional new knowledge may help improve treatment and prophylaxis.

The surface of the oropharyngeal cavity is covered with a multilayer stratified squamous epithelium supported by the lamina propria, which is a layer of fibrous connective tissue [1,2]. Stratified oropharyngeal epithelial cells from the parabasal to the granulosum layers have well-developed lateral adherens and tight junctions, indicating a polarized organization [3,4,5,6]. The lateral localization of adherens and tight junctions between neighboring cells of oral epithelium contribute to a physical barrier that protects the body from penetration by viruses and other pathogens [4,5]. 

The oropharyngeal mucosa also contains a broad population of adaptive and innate immune cells, including T and B cells, macrophages, dendritic/Langerhans cells (DC/LCs), and natural killer (NK) cells that are distributed within the epithelium and lamina propria [2,7,8,9,10,11,12,13,14,15,16]. Intraepithelial DC/LCs and macrophages in the oral mucosa are critical components of the innate immune system. These cells defend against pathogens that enter the body via the oral cavity [2,13,14,16,17,18,19,20,21,22]. Intraepithelial macrophages and DC/LCs are also antigen-presenting cells that are capable of activating an adaptive immune response. Thus, these cells may serve as “bridges” between the innate and adaptive immune systems [2,14,23,24,25]. In addition, oral mucosal epithelial cells express toll-like receptors (TLRs) 2, 3, 4, 5, 6, and 9, which are critical facilitators of innate immune responses against numerous pathogens [26,27]. 

Intraepithelial oral mucosal DC/LC and macrophages originate from peripheral blood CD14^+^ monocytes [28,29]. Recruitment of circulating monocytes into the mucosal epithelium is mediated by monocyte chemotactic protein-1 (MCP-1), MCP-2, macrophage inflammatory protein-1 alpha (MIP-1α), and MIP-1β [30,31,32]. Expression and secretion of these mediators in the mucosal epithelium are modulated by multiple chemokine/cytokines, including interferon-ɣ (IFN-ɣ), tumor necrosis factor-α (TNF-α), and interleukins (ILs), including IL-1, IL-1β, IL-4, IL-6, IL-8, IL10, IL-13, and IL-15 [14,32,33,34,35,36]. MCP-1 is secreted from the basolateral membranes of polarized epithelial cells [37,38]; the polarized release of MCP-1 may generate a gradient toward blood vessels that serve to recruit monocytes into the epithelium. Once they have transited across the endothelial layer, monocytes differentiate into macrophages and DC/LCs [11]. Further traffic of monocytes/macrophages, DCs, and T lymphocytes within the mucosal epithelium is coordinated by the formation of transient tight junctions between immune and epithelial cells [39,40,41,42]. Migrating immune cells, particularly DC/LCs, express the tight junction proteins known as claudin-1 and occludin. Transient association of these proteins with cell junctions promotes the migration of immune cells without disrupting epithelial barrier functions [39,41,43,44]. This allows DC/LCs to reach the mucosal surface [39,43]. The absence of epithelial junctions may reduce the efficiency of epithelial–lymphocyte interactions, thereby leading to dysfunction (i.e., little to no retention of interepithelial lymphocytes and DC/LCs and thus their depletion) [45,46,47]. Interactions of polarized mucosal epithelial cells with DC/LCs and other cells of the adaptive and innate immune systems are critical for maintaining oral mucosal immune homeostasis.

The oropharyngeal mucosal epithelium and its intraepithelial and subepithelial adaptive and innate immune cells play critical roles in promoting protection against numerous pathogens, including viruses, bacteria, and fungi. However, in HIV/AIDS, various chronic disorders can develop with a significant impact on the oral mucosal epithelium. These include inflammation, necrotizing mucosal ulcers, and malignant and nonmalignant lesions that may impair the barrier as well as the innate and acquired immune functions of the oral mucosa [48,49,50,51,52,53,54,55].

Systemic HIV/AIDS is accompanied by the spread of cell-free and cell-associated HIV-1 in the oral mucosal environment. There are several reports of viral DNA/RNA, cell-free HIV-1 virions, and tat and gp120 proteins isolated from oral mucosal tissue and saliva of HIV/AIDS patients [6,56,57,58,59,60,61,62,63,64]. HIV-infected DC/LCs, lymphocytes, and macrophages were also detected in the mucosal and submucosal layers of the oropharyngeal epithelium [6,56,57,62,63,65]. Electron microscopy revealed that HIV virions could be found within the tight junctions of the oral epithelium [62]. The presence of both cell-free and cell-associated HIV-1 both around and within oropharyngeal mucosa may result in numerous changes that impair its innate immune and barrier functions. 

## 2. Role of Oropharyngeal Mucosal Epithelium in HIV-1 Transmission in Adults and Children

Oral HIV-1 transmission in the adult population may occur during oral sex, in newborn children and infants during delivery and breastfeeding [66,67,68,69,70,71,72,73]. The rate of adult oral HIV-1 transmission has been estimated at ~0.004% per exposure (i.e., not a highly efficient process) [66,67,68]. By contrast, the rate of mother-to-child transmission (MTCT) of HIV-1 in the absence of antiretroviral therapy (ART) may be as high as 15% in Europe and 25–30% in Asian and African countries [74,75,76]. 

The oropharyngeal and tonsillar mucosal epithelium may express one or more co-receptors or non-canonical HIV-1 receptors that may facilitate virus binding and entry, including CC chemokine receptor type 5 (CCR5), CXC chemokine receptor type 4 (CXCR4), heparan sulfate proteoglycans (HSPGs), mannose receptor, galactosylceramide (GalCer), and T-cell immunoglobulin and mucin domain 1 (TIM-1) [3,77,78,79,80,81,82,83,84]. 

Results of studies featuring ex vivo adult and fetal/infant tissue explants revealed that HIV-1 transmission through the adult oral epithelium was less efficient than fetal/infant epithelial tissues, which supported rapid viral transmigration through the mucosal epithelium and infection of virus-susceptible intraepithelial and subepithelial cells [3,85]. The resistance provided by the adult tissues was primarily due to the presence of multiple epithelial layers and tissue stratification (20–30) with highly-effective tight junctions (Figure 1). The highly stratified adult oral epithelial cells limit viral penetration more efficiently than the less stratified fetal/neonatal/infant counterparts (i.e., with 3–5 layers) [3].

Furthermore, adult oral epithelial cells express high levels of anti-HIV-1 innate proteins, including human beta-defensin (hBD)2 and hBD3. These cells also express secretory leukocyte protease inhibitor that inactivates intraepithelial virions and reduces oral transmission of HIV-1 [79,82,85]. By contrast, fetal/infant oral epithelial cells do not express high levels of these proteins, which may be among the factors contributing to the comparatively high rate of HIV-1 MTCT [79,82,85]. HBDs tagged with the HIV-1 protein transduction domain known as Tat were delivered to HIV-1-infected infant tonsillar epithelial cells, which facilitated efficient penetration and virus inactivation [85]. PTD-mediated internalization of these proteins in infant tonsillar epithelial cells was followed by their penetration into MVBs and vacuoles containing HIV-1. PTD also promoted the fusion of HIV-containing vesicles with lysosomes which led to the degradation of gp120 and p24 and viral inactivation [85]. Ex vivo PTD-mediated internalization of hBD2 and hBD3 into tonsillar tissue explants from infants also reduced virus spread from epithelial cells to CD68^+^ macrophages, CD4^+^ T lymphocytes, and CD1c^+^ DCs [85].

HIV-1 internalization through the apical surface into infant tonsillar epithelial cells can be initiated by multiple entry pathways, including micropinocytosis as well as clathrin and caveolin/lipid raft-associated endocytosis [83]. An evaluation of HIV-1 transmission through polarized tonsillar epithelial cells revealed that approximately 0.05% of inoculated virions underwent transcytosis across the epithelium [86]. More than 90% of the internalized virions were sequestered in epithelial endosomes that included multivesicular bodies (MVBs) and vacuoles. Sequestration of HIV-1 in the endosomal compartment of tonsillar epithelial cells was observed both in the single layer of polarized cells as well as ex vivo in explants of tonsillar epithelial tissue [83,85,86]. Intraepithelial HIV-1 remained infectious for several days, although no virion release was observed [86]. Interactions of HIV-1-containing epithelial cells with activated peripheral blood mononuclear cells and CD4^+^ T lymphocytes led to the disruption of epithelial cortical actin and the spread of the virus from epithelial cells to the lymphocytes. IFN-ɣ and TNF-α treatment of tonsillar epithelial cells also induced reorganization of cortical actin and intracellular virion release [86]. 

The release of HIV-1 from oropharyngeal mucosal epithelial cells may result in the virus spreading into intraepithelial and subepithelial macrophages, DC/LCs, and CD4 T^+^ lymphocytes. This is the first step in establishing systemic HIV-1 infection. Mucosal macrophages, DC/LCs, and intraepithelial T lymphocytes, DC/LCs may then transmit HIV-1 across the mucosal epithelium into regional lymph nodes [87,88,89,90,91,92]. 

Oral transmission of HIV-1 may also result from paracellular virus penetration if the integrity of the oral mucosal epithelium is impaired. HIV-1 gp120 binding to GalCer can result in elevated levels of intracellular calcium and activation of mitogen-activated protein kinase (MAPK) and PI3K signaling [93,94,95]. In addition, HIV-1 envelope protein gp120-induced activation of MAPK and NF-κB signaling reduced the expression of ZO-1, occludin, and claudin-1 in oral epithelial cells, leading to the disruption of tight junctions [96,97,98]; this may facilitate paracellular penetration of HIV-1 virions. HIV/AIDS-associated production and release of proinflammatory cytokines, including TNF-α and IFN-γ, may also disrupt tight junctions of oral epithelial cells and lead to paracellular penetration of HIV-1 [4,5].

Oral epithelial cells may also support non-replicative HIV-1 infection and virus transfer to CD4^+^ T lymphocytes [99] and immobilize the infectious virions on their surfaces to facilitate their transfer to permissive cells [100]. Likewise, ex vivo HIV-1 infection of tonsillar explants can lead to a productive infection of intraepithelial and submucosal macrophages and lymphocytes [65,96,101,102].

## 3. Manifestations of HIV/AIDS in the Oropharyngeal Mucosal Epithelium

### 3.1. HIV/AIDS Reactivates Opportunistic Infections of the Oral Mucosa 

The oral cavity of a healthy individual maintains abundant flora, including many commensal and potentially opportunistic organisms. Inflammatory lesions and opportunistic infections are relatively infrequent, suggesting that the oral mucosa has highly efficient biological and immunological barrier functions [14,16,103,104]. The innate immune and barrier functions of the oral mucosal epithelium may become severely impaired in individuals with HIV/AIDS, which will permit the development of opportunistic infections. Herpes simplex virus-1 (HSV-1), Epstein-Barr Virus (EBV), and human cytomegalovirus (HCMV) are common oral pathogens that can lead to various chronic disorders in the oral mucosal epithelium, including necrotizing mucosal ulcers and nonmalignant lesions [48,49,50,51,52,53,105]. All three of these viruses can infect children early in life via the oral mucosa and persist throughout their lifetimes [106,107,108]. Reactivation of EBV, HSV-1, and HCMV in individuals diagnosed with HIV/AIDS leads to viral shedding into various body fluids, including saliva and cervicovaginal secretions [55,109,110,111,112]. Numerous human herpes viruses (including EBV, HSV-1, and HCMV) were detected in the saliva of HIV-infected individuals despite the availability of ART [109]. These viruses can be transmitted to neonates and infants through maternal saliva and/or breast milk [106,113,114,115,116,117]. Oropharyngeal shedding of HSV and HCMV was also reported in HIV-infected children [118,119].

#### 3.1.1. Synergistic Contributions of HIV-1 and EBV to the Nonmalignant Oral Lesion, Hairy Leukoplakia

White epithelial lesions on the side of the tongue, a condition known as hairy leukoplakia (HL), are well-known oral mucosal manifestations of HIV/AIDS. These lesions, which feature epithelial acanthosis and hyperkeratosis without inflammation [120,121,122], are associated with high-level replication of EBV [120,121,122]. HL lesions consistently demonstrate severe depletion of intraepithelial LCs, suggesting the critical role of these cells in disease pathogenesis [123,124,125]. Ex vivo experiments targeting buccal explants and tongue tissue revealed that EBV infection occurs first in submucosal CD14^+^ monocytes. The virus then spreads via migration to the epithelium. This series of events will initiate a productive EBV infection within terminally-differentiated cells of the spinosum and granulosum layers [126]. Exposure of EBV-infected oral explants or EBV-infected monocytes with specific antibodies that target CC chemokine receptor 2 (CCR2) and MCP-1 prevented monocyte entry and blocked keratinocytes infection. Interestingly, EBV-infected B-lymphocytes played only a small role in EBV spread to keratinocytes in an ex vivo oral tissue explant model [126]. However, co-cultivation experiments performed in vitro revealed that infected B-lymphocytes may promote EBV spread to previously uninfected monocytes. Circulating EBV-positive monocytes are present in most HIV-infected individuals. This finding is consistent with a pathway in which EBV spreads from B-lymphocytes to monocytes. Infected monocytes entering the epithelium will then differentiate into macrophages and/or LCs. These cells will elicit a productive keratinocyte infection and thus promote the development of the characteristic HL lesions [126] (Figure 2). It is also possible that CD14-negative precursors of LCs may disseminate EBV within oral mucosal epithelial cells [127].

Both EBV and HIV-1 can infect macrophages, monocytes, and DCs [65,128,129,130,131,132,133,134]. Circulating monocytes serve as reservoirs for HIV-1 infection [135]. Circulating monocytes in patients undergoing highly active ART (HAART) may harbor replication-competent, non-latent HIV-1 [128]. EBV-infected macrophages were also detected in healthy asymptomatic individuals. This finding suggested that macrophages may be a reservoir for this opportunistic pathogen [136].

HIV-1 infection of monocytes may increase the surface transport of adhesion molecules and integrins, including the α5β1 and αvβ3 integrins [137,138,139,140,141]. The EBV envelope protein BMRF-2 uses β1 and α5β1 integrins as receptors to infect polarized oral epithelial cells [142,143]. EBV entry into monocytes may also be mediated by an EBV BMRF-2– integrin interaction because monocytes do not express the primary EBV receptor, CD21 [144]. Thus, HIV/AIDS-associated activation of surface integrin expression in peripheral blood monocytes may facilitate their infection by EBV. In addition, the HIV tat protein induces endothelial cell expression of intercellular adhesion molecule (ICAM), vascular cell adhesion molecule-1 (VCAM-1), and endothelial leukocyte adhesion molecule (E-selectin) [145]. The synergy between HIV tat protein and TNF-α detected in elevated levels in association with HIV/AIDS increases β2 integrin expression in monocytes; these responses may lead to active translocation of monocytes from the circulation into tissue sites where they differentiate into LCs [146,147]. This may lead to the migration of HIV-1 and/or EBV-infected monocytes from the circulation into mucosal sites, thereby initiating EBV spread from LCs to differentiated epithelial cells capable of supporting productive virus replication [126,127]. EBV BMRF-2/BDLF-2 facilitates virus spread between oral epithelial cells and establishes foci of virus-infected epithelial tissue [148,149]. This may promote epithelial cell proliferation and lead to the development of HL. EBV BHRF-1 delays or inhibits apoptosis of cells in the differentiated granulosum layers, which may permit enhanced proliferation of epithelial cells, thereby increasing the thickness and hyperplasia characteristic of the HL epithelium [150].

HIV-1 infection of monocytes/macrophages and DC/LCs may impair their critical functions, including activation, maturation, antigen presentation, and interactions with T cells [151,152,153,154,155]. HIV and/or EBV infection of DC/LCs may ultimately result in their dysfunction. This may lead to the depletion of oral mucosal intraepithelial LCs, as has been shown for HIV/AIDS-associated HL lesions [123,124,125,156]. HIV-1 gp120 and tat proteins are also capable of disrupting epithelial adherens and tight junctions [96,97,98]. This may reduce the intraepithelial retention of LCs and lead to local depletion [39,41,43,44,157].

ART for HIV-1 and treatments effective against EBV typically lead to the resolution of HL in most patients [124,158]. These findings indicate the synergistic roles of HIV and EBV in the development of HL. However, although the introduction of ART has reduced the frequency of HL, this lesion has not been completely eliminated [159,160]. This suggests that residual HIV-associated mechanisms may contribute to HL development. 

#### 3.1.2. HIV-Induced Disruption of Tight and Adherens Junctions of Oral Epithelial Cells Facilitates the Spread of HSV-1

Herpes simplex virus (HSV) type 1 and HSV type 2 are both opportunistic infections that are frequently associated with HIV/AIDS [105]. HSV-1 reactivates and replicates in the oral epithelium of HIV/AIDS-associated immunocompromised individuals and can lead to oral ulcers, gingivitis, and necrotic lesions [55]. HSV-1 reactivation may occur despite ongoing HAART [161,162]. Although the increased risk of developing HSV infection may be mediated in part by HIV-induced immune dysfunction, they may also be associated with direct and/or indirect molecular interactions between these two viruses. 

HIV-infected LC/DCs, CD4^+^ T lymphocytes, and macrophages infiltrating the mucosal epithelium can release virions as well as the viral proteins, gp120 and tat. Interactions of virions and viral proteins with one or more HIV-coreceptors (i.e., CXCR4, CCR5, HSPG, and/or GalCer), TLR-2/4, integrins, and/or mannose receptors on epithelial cells induces the activation of PI3K, MAPK, TLR, and NF-κB signaling pathways, resulting in the expression of proinflammatory cytokines (TNF-α, IFN-ɣ, IL-1, IL-1β, IL-2, IL-6, IL-8, and IL-13), matrix metalloproteinases (MMPs-2 and -9), and caspases (caspase-3 and -6) [6,98,163,164,165]. In addition, activation of these pathways will downregulate the expression and/or promote aberrant internalization and degradation of epithelial adherens and tight junctions.

Prolonged interactions of polarized epithelial cells from the oral cavity with tat and gp120 can lead to the disruption of adherens and tight junctions via activation of the MAPK signaling pathway [97]. HIV-associated activation of MAPK leads to upregulation of MMP-9 and NF-κB which induces the disruption of adherens and tight junctions [98]. Furthermore, HIV-induced disruption of oral epithelial junctions facilitates the paracellular spread of HSV-1 [97] (Figure 3). 

The HSV-1 envelope glycoprotein gD binds to nectin-1, which is a cell adhesion protein [166] sequestered within the intercellular junctions and limits HSV access to mucosal epithelial cells [167]. HIV-mediated disruption of adherens junctions will liberate nectin-1, thereby promoting HSV-1 binding to gD. HSV-1 infection is substantially increased in epithelial cells whose junctions have been disrupted compared to those with junctions that remain intact [97]. Exposure of nectin-1 due to disruptions in adherens junctions may also accelerate the cell-to-cell spread of HSV-1 from oral epithelial cells infected to those uninfected. Exposure to anti-nectin-1 and anti-HSV-1 gD antibodies will result in a substantial reduction in the extent of HSV-1 infection as well as cell-to-cell spread. Collectively, these findings suggest that HIV-1-mediated HSV spread and ongoing infection are facilitated by HSV gD interactions with exposed nectin-1 [97]. 

HSV is latent in sensory neurons in healthy individuals with intact immune function [168]. HIV/AIDS may induce HSV reactivation in neurons, thereby leading to the development of mucosal disorders [161]. In addition, HIV-mediated release of nectin-1 from cells with disrupted adherens junctions may also promote the spread of HSV-1 from neurons to epithelial cells (Figure 3). Thus, HIV-mediated disruption of intercellular junctions potentiates HSV-1 infection as well as cell-to-cell and paracellular spread within the oral mucosal epithelium. This mechanism may contribute to the rapid development of HSV-associated oral lesions in HIV-infected individuals.

#### 3.1.3. HIV-1 and HCMV Mother-to-Child Transmission (MTCT) through the Infant tonsillar Epithelium Synergistically Promote the Spread of Both HIV-1 and HCMV

Despite the availability of effective ART, 100,000–240,000 cases of MTCT of HIV-1 are reported each year [76,169,170,171,172]. Most cases of MTCT (40–50%) result from breastfeeding [76]. Breast milk may contain cell-free and cell-associated HIV-1 [69,70,71,72,73], which may initiate mucosal transmission of the virus via the child’s oropharyngeal and gastrointestinal mucosae [3,6,79,173]. Breast milk may also contain HCMV that has been activated in response to HIV/AIDS [174], which can promote the development of oral mucosal lesions, retinitis, hepatitis, esophagitis, pneumonia, encephalopathy, and/or gastrointestinal inflammation [175,176,177,178,179]. Nearly all HIV-infected individuals, including pregnant women, exhibit HCMV co-infection [108,180]. MTCT of HCMV can occur in utero via the placenta and during labor by exposure to secretions from the cervix and the vagina [119,181]. However, most HCMV MTCT is observed postpartum by virion transfer in infected breast milk and transmission through the oral and/or gut mucosal epithelium [119,181,182]. Almost all cases of HCMV activation in HCMV-seropositive, HIV-uninfected women occur secondary to lactation [183]. The virus can be found in breast milk for only a few weeks postpartum [184,185]. By contrast, HCMV shedding into breast milk of HIV-infected women may increase significantly during the postpartum period and can last for six months or longer. Persistent viral shedding can facilitate MTCT via the infant’s oropharyngeal and gastrointestinal mucosal epithelium [186]. Low maternal CD4^+^ T cell counts and high levels of HIV RNA in breast milk correlate with comparatively high levels of HCMV genomic DNA in breast milk [186,187]. HCMV transmission (perinatal and postnatal) may occur in as many as 90% of children who acquire the virus during early childhood [119,176,188,189]. Primary HCMV infection or disease reactivation in HIV-infected pregnant women may also contribute to the MTCT of HIV-1. Several groups have shown that elevated levels of HCMV in breast milk are associated with an increased risk of postpartum MTCT of HIV-1 [186,189,190,191]. 

Exposure to cell-free forms of HIV-1 or HIV-1 proteins gp120 and tat will lead to the disruption of the tight junctions characteristic of epithelial cells from the tonsils. This will also increase paracellular transfer through polarized epithelial cells from infant tonsils as well as ex vivo tissue explants. This will also facilitate HCMV spread within the tonsil epithelial cells [96]. Furthermore, HIV-1 gp120 and tat-induced activation of NF-κB and MAPK signaling in these cells increases the extent of HCMV infection. NF-κB activation is a requirement for transactivation of the major immediate early HCMV promoter and promotes the expression of viral IE proteins [192,193,194]. By contrast, MAPK activation is required for HCMV replication and the generation of viral progeny [195].

HCMV infection of epithelial cells from human tonsils may also result in the disruption of tight junctions. This will increase paracellular transfer and thus facilitate HIV-1 spread into the mucosal layers [96] (Figure 4). HCMV-induced paracellular spread of HIV-1 in tonsil tissue also enhances virus access to macrophages, dendritic cells, and CD4^+^ T lymphocytes. In addition, HIV-1-enhanced HCMV infection via paracellular spread to epithelial cells will also lead to HCMV infection of tonsillar DCs and macrophages. Thus, HIV-1- and HCMV-mediated disruption of the tight junctions of the infant tonsillar mucosal epithelial cell tight junctions may impair their barrier function, thereby facilitating paracellular penetration and the initiation of MTCT (Figure 4). 

HIV-1- and HCMV-co-infected tonsillar explant tissues showed higher levels of target cell infection with both viruses than was observed in tissues infected independently with either virus alone [96]. These results highlight the synergistic effects of both HIV-1 and HCMV with respect to promoting infection. This synergism may result in critical contributions to HIV-1 and HCMV transmission via breast milk when both viruses are present [186,187] and can interact simultaneously with the infant/newborn oral mucosa.

## 4. HIV-1 Proteins gp120 and Tat Promote Invasiveness of Both Human Papillomavirus (HPV)-Positive and HPV-Negative Neoplastic Oral and Genital Epithelial Cells 

HIV/AIDS may increase the risk of developing cancer. Results from recent studies have revealed that the incidence of HPV-associated oropharyngeal, cervical, and anal cancer is 6, 22, and 80 times higher, respectively, in HIV-infected compared to HIV-uninfected individuals [196,197,198]. HPV is the etiological agent of most oral-genital epithelial carcinomas [199,200]. Although anti-HPV vaccination strategies are effective in preventing virus infection, the value of this approach is limited in people living with HIV (PLWH) because most of these individuals have experienced multiple HPV exposures. HIV-1 may also be a risk factor associated with head and neck as well as HPV-negative oral cancer [201,202,203,204,205]. 

HIV may increase the risk of developing HPV-associated cancers by attenuating local and systemic immune responses. Moreover, HIV-associated epithelial disruption may promote HPV infection of oral and genital epithelial cells [6]. The addition of HPV-16 pseudovirions (PsVs) to the mucosal surfaces of disrupted tissues led to their paracellular penetration into the epithelium [6]. HPV-16 PsV entry into the basal/parabasal cells (i.e., the site of initiation of the HPV life cycle) was observed. These findings suggest that HIV-associated disruption of mucosal epithelial junctions facilitates paracellular spread by oncogenic HPV virions through strata spinosum and granulosum layers, leading to the infection of basal/parabasal cells. Initiation of this infection can lead to HPV-associated neoplasia. 

HIV-1-induced disruptions of the tight and adherens junctions in mucosal epithelial cells may also lead to epithelial-mesenchymal transition (EMT) [206]. EMT, which is a physiologic process, regulates the development of cell lineage identity and cell differentiation associated with embryonic development [207]. EMT also promotes neoplasia, including the growth and metastasis of invasive epithelial cancers [208]. EMT associated with the development of epithelial cell cancer involves multiple steps, beginning with the loss of apicobasal polarity. This event may be followed by loss of adherens and tight junctions as well as critical cell-adhesive properties. Cells in transition develop a spindle cell-type shape. During the final stages of EMT, these cells will also express mesenchymal markers [209,210,211,212,213,214]. While in the intermediate stages of EMT, the cells may express both mesenchymal (i.e., vimentin) and epithelial (i.e., E-cadherin) markers, which are critical factors contributing to cancer cell invasiveness [209,210,211,212,213,214]. 

The dominant network involving the transforming growth factor (TGF)-β signaling pathway regulates EMT leading to cancer [215]. The mature form of TGF-β binds to TGFβ−R2 and promotes a signaling cascade leading to Smad family transcription factor complex activation. These events lead to the activation of several specific transcriptional regulators, including Slug, Twist1, and Snail. Activation will be followed by the upregulation of N-cadherin, fibronectin, and vimentin and the downregulation of E-cadherin [216,217]. 

Prolonged interactions (five to seven days) with HPV-16-immortalized anal AKC-2 and cervical CaSki epithelial cells with cell-free-HIV-1 virions and/or viral proteins gp120 and tat leads to the development of increased invasiveness and EMT [218] (Figure 5). Furthermore, adding these viral proteins to cultures of oral epithelial cells, including HPV-16-infected SCC-47 and HPV-16-negative HSC-3 cells, resulted in similar responses. While gp120- and tat-induced EMT resulted in the detachment of poorly-adherent cells, these same cells could undergo reattachment under appropriate conditions. Reattached cells co-expressed vimentin and E-cadherin. This result suggested that these cells had reached the intermediate stage of EMT [218]. Reattached cells also expressed stem cell markers CD44 and CD133 that may contribute to cancer-associated metastasis and invasion. Once E-cadherin expression was restored, and vimentin expression was inhibited, the HIV-induced EMT was reduced, as well as the invasiveness of HPV-16 immortalized anal epithelial and cervical cells [218]. 

Taken together, these results suggest that direct interactions of extracellular HIV-1 virions and/or gp120 and tat with neoplastic genital or oral mucosal epithelial cells may lead to the development of an EMT phenotype that accelerates the development of HPV-associated malignancies (Figure 5). Recent findings also suggest that the HIV-1 tat protein contributes to the induction of EMT and the increased invasiveness of HPV-negative cancer cells derived from the lung epithelium [219]. In addition, other studies have highlighted the impact of HIV-associated reductions in adherens and tight junction protein expression in both lung and intestinal epithelial cells [220,221]. Collectively, these results suggest that HIV-1 infection may promote the development of malignancy in both HPV-negative and HPV-infected neoplastic epithelial cells. 

## 5. Conclusions

The multi-layer adult oral mucosal epithelium expresses high levels of anti-HIV innate immune proteins hBD2, hBD3, and secretory leukocyte protease inhibitors that inactivate the virus and thus reduce the rate of oral transmission. By contrast, the oral mucosal epithelia of newborns and infants cannot efficiently prevent HIV-1 penetration because of its minimal stratification and inability to produce sufficient innate immune proteins. 

HIV-1 infection leads to EBV infection of circulating monocytes, followed by their migration into the oral mucosal epithelium and differentiation into DC/LCs. The spread of EBV from macrophages and LCs into differentiated oral epithelial cells may lead to productive viral replication and extensive cell-to-cell spread of virus progeny within the stratum granulosum of the oral mucosal epithelium, thereby initiating the development of HL. 

MAPK activation in oral epithelial cells by HIV-1 gp120 and tat leads to the upregulation of NF-κB and MMP-9, thereby disrupting tight and adherens junctions and facilitating HSV-1 paracellular spread. In addition, HIV-induced disruption of epithelial junctions also liberates sequestered nectin-1 and thus facilitates its binding to HSV-1 gD, which promotes HSV-1 spread within the epithelial cells. 

Disruption of infant tonsil epithelial tight junctions by HIV-1 and HCMV may impair the barrier function of mucosal epithelium and permit paracellular penetration of both viruses and the initiation of MTCT. 

Extended interactions between HIV-1 cell-free virions or the HIV-1 proteins gp120 and tat with HPV-positive and -negative oral neoplastic epithelial cells increase their EMT phenotype and invasiveness. 

Altogether, these studies show the molecular mechanism of the interaction between HIV-1 and other pathogens, including EBV, HCMV, HSV-1, and HPV, in the oropharyngeal mucosal epithelium. These interactions impair the barrier and innate immune functions of oropharyngeal epithelium, leading to the increased spread of these viruses within the oral mucosal environment and the progression of existing neoplastic processes of epithelial cells. Additional research may help improve treatment and prophylaxis for HIV/AIDS infection. New approaches could eliminate/reduce the reactivation and spread of opportunistic infections in the oral cavity and maintain the barrier and innate immune functions of the oropharyngeal mucosal epithelium

## Figures and Tables

**Figure 1 biomedicines-11-01444-f001:**
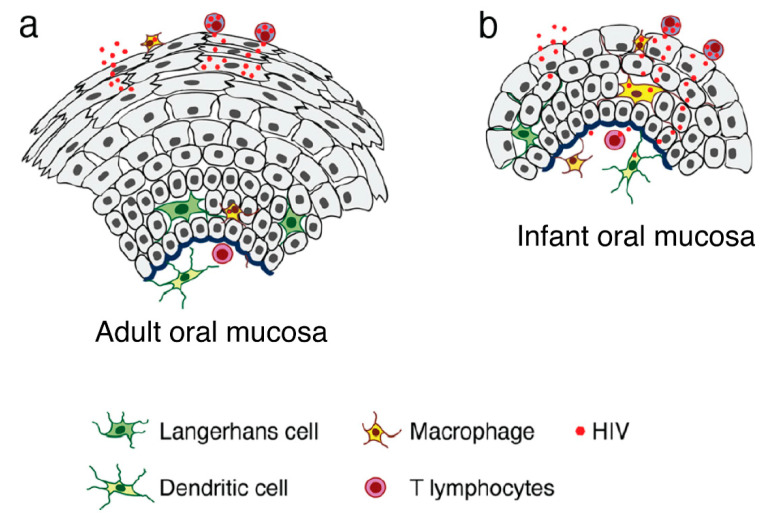
Model of HIV transmigration in the infant/fetal and adult oral epithelium. (**a**) Adult oral epithelial cells are stratified into numerous layers. While cell-free HIV can transmigrate to some extent across the upper regions of the intact adult oral epithelium (across two to five layers), the virus is unable to spread to the lower layers or the lamina propria. (**b**) By contrast, the infant/fetal oral mucosal epithelium contains only two to five epithelial layers and is not completely stratified. The spread of HIV-infected macrophages and cell-free virions across infant/fetal oral epithelial cells may result in the infection of HIV-susceptible epithelial and submucosal cells, including macrophages, LC/DCs, and T lymphocytes. Thus, fetal/infant oropharyngeal epithelial cells may serve as a critical portal for HIV entry during MTCT. Increased rates of HIV transmission across the fetal/infant oral epithelium compared to that of adults may represent both reduced barrier function (associated with pauci-stratification) as well as lower levels of innate immune proteins. The models in this and other figures were created by the authors using Adobe^®^ Illustrator (San Jose, CA, USA).

**Figure 2 biomedicines-11-01444-f002:**
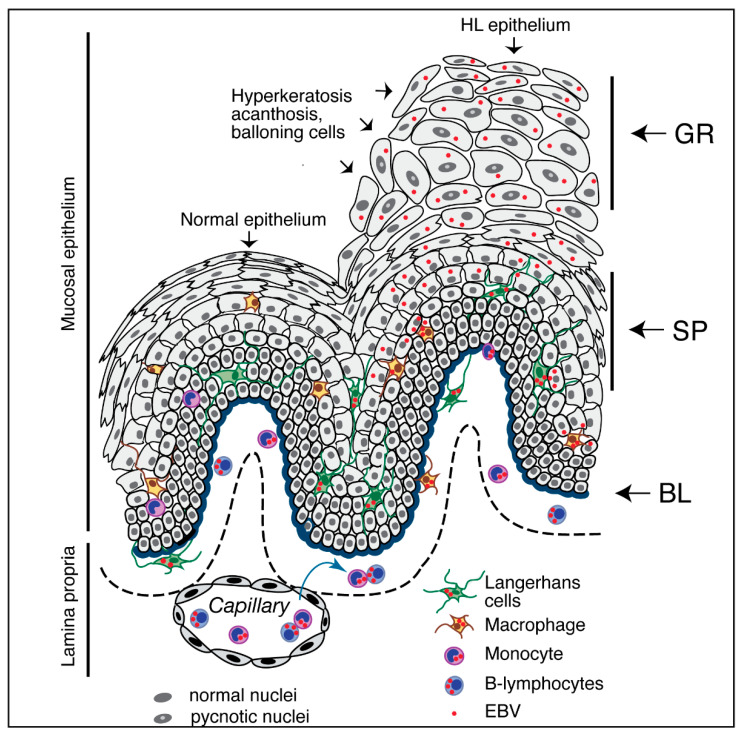
Role of HIV-1 and EBV infection in stratified oral mucosal epithelial cells and development of HL. Latent EBV may be reactivated in immunocompromised hosts. This response can be seen in individuals with immunodeficiency secondary to HIV and will lead to virus replication in B-lymphocytes. Infectious virions that are generated in and released from B-lymphocytes will then infect peripheral blood monocytes and/or cells in the lamina propria of the mucosal epithelium in the oral cavity; the monocytes will then become EBV-positive macrophages and LCs. EBV-infected LCs and macrophages will migrate into the mucosal epithelium, where they will transmit the virus to epithelial cells in the spinosum layer. The terminally-differentiated epithelial cells in this layer support lytic EBV infection and can produce and release large amounts of infectious progeny virions. Productive EBV infection in this locale may also result in virus spread to the upper granulosum layers of oral epithelium and thus to the development of HL, with characteristic acanthosis, hyperkeratosis, and ballooning morphology. Symbols in the figure include GR—stratum granulosum; SP—stratum spinosum; BL—stratum basale.

**Figure 3 biomedicines-11-01444-f003:**
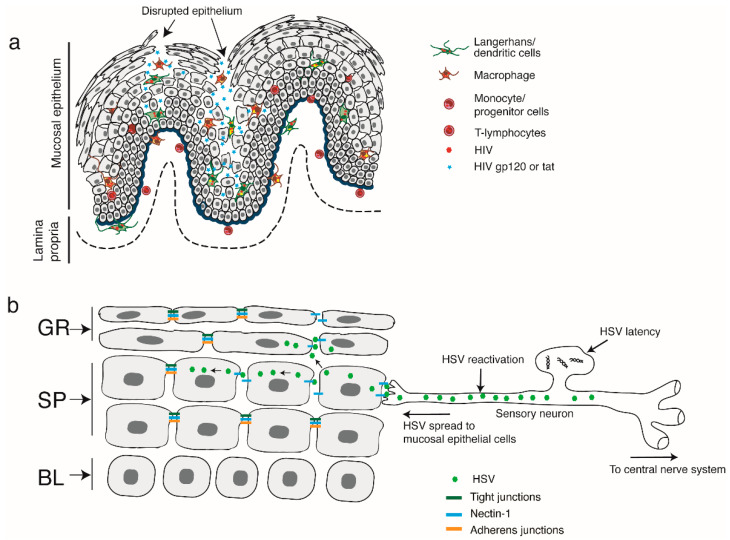
HIV/AIDS-associated disruption of mucosal epithelium facilitates the spread of HSV infection. (**a**) HIV-infected LC/DCs, macrophages, and CD4^+^ T lymphocytes infiltrate the mucosal epithelium, where they release virions as well as gp120 and tat. Viruses and viral proteins mediate the disruption of epithelial adherens and tight junctions, thereby impairing epithelial integrity. (**b**) HIV-associated immune dysfunction leads to the reactivation of HSV-1 in sensory neurons. The reactivated virus infects epithelial cells via interactions of viral envelope glycoprotein D with its receptor known as nectin-1. Nectin-1 is sequestered within the regions associated with intact adherens junctions in the lateral membranes of epithelial cells. HIV-induced disruption of adherens junctions liberates sequestered nectin-1, thereby facilitating its interactions with HSV glycoprotein D. This promotes HSV infection and cell-to-cell spread from neurons to epithelial cells as well as between cells in the oral epithelium. Virus spread leads to the rapid progression of HSV-mediated mucosal lesions and ulcers. Symbols in the figure include GR—granulosum; SP—spinosum; BL—basal.

**Figure 4 biomedicines-11-01444-f004:**
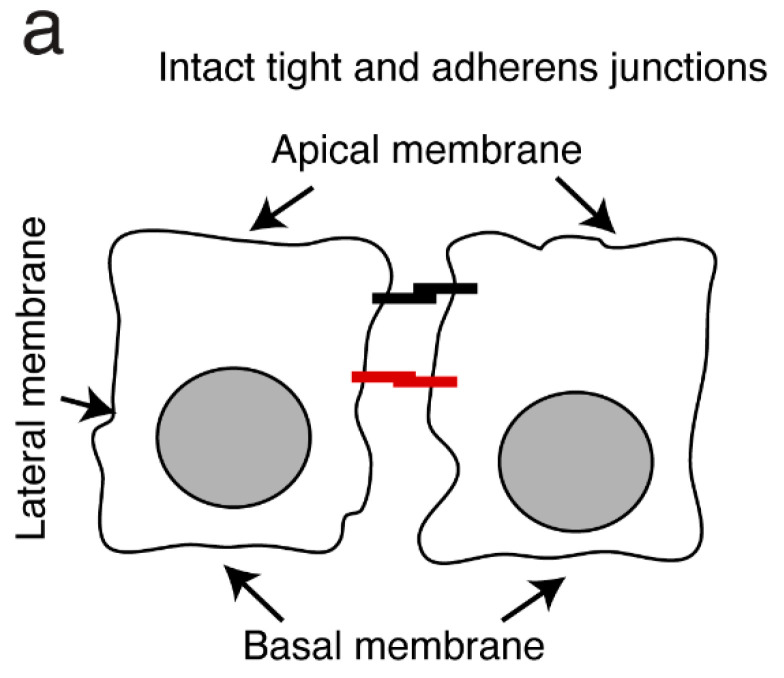
Co-infection of tonsil tissues with HIV-1 and HCMV may amplify both viral infections in a synergistic fashion [96]. (**a**) Tonsillar and oropharyngeal epithelial cells have well-developed tight and adherens junctions that contribute to the barrier preventing the paracellular spread of HCMV and HIV-1. (**b**) Most MTCT of these viruses occurs during childbirth and breastfeeding. At these times, the viruses may have the opportunity for simultaneous interactions with oral mucosal epithelial cells and can thereby disrupt their integrity. Cell-free HIV-1 and viral gp120 found in breast milk and cervicovaginal secretions may promote the disruption of junctions in epithelial cells in the oropharynx and the tonsils. Secreted tat protein may also promote disruption of tight junctions in HIV-infected infants. Collectively, these interactions may promote MTCT of HCMV and may also amplify its paracellular spread. HCMV infection of the mucosal epithelial cells of the oropharynx also enhances the paracellular spread of HIV-1 and initiates HIV-1 MTCT. (**c**) Virus-mediated disruption of the integrity of the epithelial mucosa may promote HIV-1 paracellular spread by providing increased access to intramucosal and submucosal DCs, macrophages, and CD4^+^ T lymphocytes. Virus-induced mucosal epithelial disruption may also promote infection via HCMV paracellular spread. HCMV can then spread to intraepithelial and submucosal DCs and macrophages. Thus, HIV-1 and HCMV may act synergistically to promote MTCT through the infant tonsillar mucosal epithelium.

**Figure 5 biomedicines-11-01444-f005:**
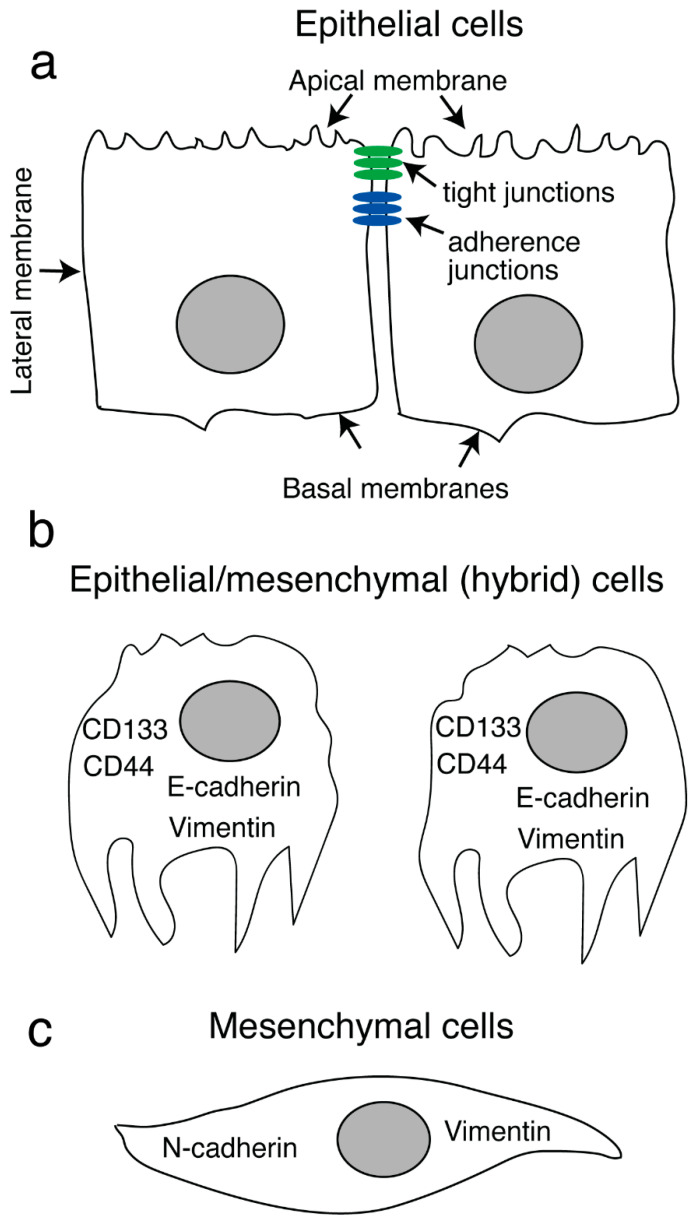
A model of EMT induced by HIV [218]. (**a**) Oropharyngeal and genital epithelial cells display a polarized organization with distinct basolateral and apical membranes as well as well-developed adherens and tight junctions. (**b**) Cell-free HIV-1 virions and gp120 and tat interact with epithelial cells, which leads to the activation of TGF-β and MAPK signaling and the induction of an EMT phenotype. As part of this phenotype, epithelial cells lose adherens and tight junctions and apicobasal polarity, as well as their cobblestone-like morphology. These epithelial cells also express vimentin and E-cadherin. Cells exhibiting this phenotype may also express markers of stem cells and thus may undergo conversion to cancer stem cells that are capable of generating invasive metastatic disease by migrating into systemic circulation through basement membranes. (**c**) Additional progression of EMT may result in upregulated N-cadherin and vimentin expression associated with loss of E-cadherin in association with the aforementioned spindle-like morphology. These mesenchymal cells are highly invasive and capable of migration. Thus, this transition may ultimately promote the spread of malignant cells within tissues and organs. Collectively, these results permit us to create a model of accelerated HIV-associated neoplasia that features both premalignant and malignant HPV-uninfected and HPV-infected mucosal epithelial cells.

## Data Availability

No new data were created or analyzed in this study. Data sharing is not applicable to this article.

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
