# Peer review of "Molecular Pathogenesis of Human Immunodeficiency Virus-Associated Disease of Oropharyngeal Mucosal Epithelium"

_biomedicines, 2023, doi:10.3390/biomedicines11051444_

Round 1
Reviewer 1 Report
Dear editor ,
I reviewed the manuscript by Tugizov entitled “ Molecular pathogenesis of human immunodeficiency virus associated disease of oropharyngeal mucosal epithelium” the manuscript is well written and may provide interesting information for the reader or the Journal.
Accordingly, the manuscript can be published in the present form.
Many thanks
Author Response
Reviewer 1
I reviewed the manuscript by Tugizov entitled “Molecular pathogenesis of human immunodeficiency virus associated disease of oropharyngeal mucosal epithelium” the manuscript is well written and may provide interesting information for the reader or the Journal.
Accordingly, the manuscript can be published in the present form.
Response: Many thanks to Reviewer 1 for reviewing this paper and for the positive feedback.
Reviewer 2 Report
Dear author,
It is an interesting review focusing on a major topic that is the molecular pathogenesis of human immunodeficiency virus-associated disease of oropharyngeal mucosal epithelium. This study focusses on the difference between children and adults. The manuscript is well written and very informative. The figures are very important.
However, several comments should be considered:
Abstract
1. It should indicate results from section 2
Introduction
1. The objective of this review should be indicated
Conclusion
1. The conclusion is very long and should be shortened
Author Response
Reviewer 2
It is an interesting review focusing on a major topic that is the molecular pathogenesis of human immunodeficiency virus-associated disease of oropharyngeal mucosal epithelium. This study focusses on the difference between children and adults. The manuscript is well written and very informative. The figures are very important.
However, several comments should be considered:
Abstract
- It should indicate results from section 2
Response: In the revised version of the paper, the results from Section 2 are included in the Abstract.
Introduction
- The objective of this review should be indicated
Response: In the revised version of the paper, we have stated the objective of the review.
Conclusion
- The conclusion is very long and should be shortened
Response: The conclusion has been shortened considerably.
Reviewer 3 Report
Dear authors,
It was a pleasure to review the manuscript "Molecular pathogenesis of human immunodeficiency virus-as-2 sociated disease of oropharyngeal mucosal epithelium". I find the flow of the writing quite adequate and explanatory. I would only like to stress the following points.
1. I believe the first 4-5 sentences of the abstract should be part of the intro and maybe not here. Also, I don't see any info about section 2 (comparing mucosal epithelium and spread of HIV between adult and infants)
2. line 81-85 The first paragraph here should be part of the intro and it is not related to the title of the section. For example it should start like "studies have shown that there is a difference in the transmission of HIV between adults and children" etc
3. line 177-186 I don't understand how the title and the content of this paragraph is related to the rest info of section 3. The rest of the section 3 talks about the interactions of HIV with other viruses such as EBV, HSV. Maybe this paragraph should be part of the intro.
4. line 526-532 I believe it should write a sentence about the invasiveness of both human papillomavirus (HPV)-positive and HPV-negative neoplastic oral and genital epithelial cells.
Also I don't see a closing sentence and a final conclusion.
Author Response
Reviewer 3
This is extensive work, with many quality images, but which source is not provided.
Response: All illustrations were created by the author using Adobe® Illustrator and are presented as a model to explain and summarize the results presented in the publications reviewed.
The entire manuscript, as well as the reference list are too long.
Response: The author was invited to write a review article to commemorate 40 years of research since the onset of the HIV/AIDS pandemic with a specific focus on findings that covered the manifestation of HIV/AIDS in the oropharyngeal mucosal epithelium. Given the long period of time that needed to be covered, the paper will by definition be somewhat lengthy and will cite many primary sources. The length of the text and the number of references does not exceed the journal specifications for a review article.
However, this material seems to be rather a course than a paper suitable for a journal.
Response: This article reviews our current understanding of the manifestation of HIV/AIDS in the oropharyngeal mucosal epithelium. While the materials presented in this paper could be adapted for use in courses focused on Oral Biology and Virology, this feature does not in any way preclude its publication as a review article in Biomedicines.
- HIV/AIDS disease remains an unsolved problem; additional new knowledge may help to improve treatment and prophylaxis.
Response: We agree with and thank Reviewer 2 for highlighting this important aspect of HIV/AIDS disease.
- The topic is relevant; I am not aware as much as it is original. It sure addresses a specific gap, among which the role of oropharyngeal mucosal epithelium in HIV-1 transmission in adults and children, manifestations of HIV/AIDS in the oropharyngeal mucosal epithelium. However, the pages 1-4 do not provide original elements, they are more or less a state of the art of the problem.
Response: The goal of this review article was to describe and summarize the past 40 years of research findings focused on the manifestations of HIV/AIDS in the oropharyngeal mucosal epithelium. As this is an unsolved problem (as Reviewer #2 notes above), the topic is both relevant and original.
Figure 1 -Provides information about the model of HIV transmigration in adult and children oral epithelial cells, but did the author paint the image by himself?
Response: Yes, this original schematic illustration was designed by the author using Adobe® Illustrator.
Figure 2 - The role of HIV-1 and EBV infection in stratified oral mucosal epithelium and development of HL - also is very suggestive, but is this the original source?
Response: This is a schematic model of HIV-1 and EBV infection in the stratified oral mucosal epithelium (Figure 2) was also designed by the author using Adobe® Illustrator.
Figure 3. - HIV/AIDS-associated disruption of mucosal epithelium facilitates spread of HSV infection and Figure 4. HIV-1 and HCMV co-infection of tonsil tissues - the same issue
Response: These schematic models of HIV-1, HSV, and HIV-HCMV spread in the stratified oral mucosal epithelium were also designed by the author using Adobe® Illustrator.
- I am not sure/aware if it contains new material.
Response: This review article covered the past 40 years of research focused on the manifestations of HIV/AIDS in the oropharyngeal mucosal epithelium. The review highlighted published materials that include both old and new findings.
- The whole paper is not organized as an: introduction, material and method, results, discussion, and conclusion, hence it is difficult to provide a specific workflow.
Response: This was written according to the instructions provided by Biomedicines for review papers. This type of paper does not have materials and methods, results, and discussion sections.
- The conclusions are generalities about the oral epithelium and HIV infection, they are a repetition of the provided information.
Response: The conclusions reflect the current state-of-the-art on manifestations of HIV/AIDS in the oropharyngeal mucosal epithelium.
The conclusion is too long.
Response: In the revised paper, the conclusion section has been shortened substantially.
- The references are too many (a total of 223!) and contain a lot of the author's self-work: 3-6, 58, 61-64 , 66, 88-89, 127, 143-144,149, 174.
Response: This review covers 40 years of research findings that document the manifestations of HIV/AIDS in the oropharyngeal mucosal epithelium. We attempted to include all critical references associated with this field. As the author’s laboratory also studies the mechanisms of HIV and HIV-associated viral infection in oral epithelium, papers from our laboratory were of course included. Nevertheless, the number of references did not exceed the limitations set by the journals.
- The figures seem to be the model of a course/educational material. I am not aware if they are original.
Response: All figures are original and designed by the author using Adobe® Illustrator. They could certainly be used as course/educational material after publication.
Reviewer 4 Report
Dear Author,
This is extensive work, with many quality images, but which source is not provided.
The entire manuscript, as well as the reference list are too long.
However, this material seems to be rather a course than a paper suitable for a journal.
1. HIV/AIDS disease remains an unsolved problem; additional new knowledge may help to improve treatment and prophylaxis
2. The topic is relevant, I am not aware as musch as it is original. It sure addresses a specific gap, among which the role of oropharyngeal mucosal epithelium in HIV-1 transmission in adults and children, manifestations of HIV/AIDS in the oropharyngeal mucosal epithelium.
However, the pages 1-4 do not provide original elements, they are more or less a state of the art of the problem.
Figure 1 -Provides information about the model of HIV transmigration in adult and children oral epithelial cells, but did the author paint the image by himself?
Figure 2 - The role of HIV-1 and EBV infection in stratified oral mucosal epithelium and development of HL - also is very suggestive, but is this the original source?
Figure 3. - HIV/AIDS-associated disruption of mucosal epithelium facilitates spread of HSV infection and Figure 4. HIV-1 and HCMV co-infection of tonsil tissues - the same issue
3. I am not sure/aware if it contains new material.
4. The whole paper is not organized as an: introduction, material and method, results, discussion, and conclusion, hence it is difficult to provide a specific workflow.
5. The conclusions are generalities about the oral epitelium and HIV infection, they are a repetition of the provided information.
The conclusion is too long.
6. The references are too many (a total of 223!) and contain a lot of the author's self-work: 3-6, 58, 61-64 , 66, 88-89, 127, 143-144,149, 174.
7. The figures seem to be the model of a course/educational material. I am not aware if they are original.
Author Response
Reviewer 4
It was a pleasure to review the manuscript "Molecular pathogenesis of human immunodeficiency virus-as-2 sociated disease of oropharyngeal mucosal epithelium". I find the flow of the writing quite adequate and explanatory. I would only like to stress the following points.
- I believe the first 4-5 sentences of the abstract should be part of the intro and maybe not here. Also, I don't see any info about section 2 (comparing mucosal epithelium and spread of HIV between adult and infants)
Response: In the revised version of the paper, the first 4-5 sentences of the abstract were moved into introduction and the results from Section 2 are included in the Abstract.
- line 81-85 The first paragraph here should be part of the intro and it is not related to the title of the section. For example, it should start like "studies have shown that there is a difference in the transmission of HIV between adults and children" etc
Response: In the revised version of the paper, this section was modified and improved as the reviewer suggested.
- line 177-186 I don't understand how the title and the content of this paragraph is related to the rest info of section 3. The rest of the section 3 talks about the interactions of HIV with other viruses such as EBV, HSV. Maybe this paragraph should be part of the intro.
Response: In the revised version of the paper, this section was modified, and the first paragraph was moved into the introduction section as the reviewer suggested.
- line 526-532 I believe it should write a sentence about the invasiveness of both human papillomavirus (HPV)-positive and HPV-negative neoplastic oral and genital epithelial cells.
Response: Yes, this paragraph was modified as suggested by the Reviewer.
- Also, I don't see a closing sentence and a final conclusion.
Response: In the revised version of the paper, we added a closing sentence and a final conclusion.
Round 2
Reviewer 4 Report
Congratulations on your work!